# Real-time monitoring of oncolytic VSV properties in a novel *in vitro* microphysiological system containing 3D multicellular tumor spheroids

**Kyoung Jin Lee[1,2]☯, Sang Woo Lee[3]☯, Ha-Na Woo[1,2], Hae Mi Cho[1,2], Dae Bong Yu[1,4], Soo Yeon Jeong[3], Chul Hyun Joo[1,2], Gi Seok Jeong[3,5]\*, Heuiran Lee[1,2]\***

1 Department of Microbiology, University of Ulsan College of Medicine, Seoul, Korea, 2 Bio-Medical Institute of Technology, University of Ulsan College of Medicine, Seoul, Korea, 3 Biomedical Engineering Research Center, Asan Institute for Life Sciences, Asan Medical Center, Seoul, Korea, 4 Department of Medical Science, Asan Medical Institute of Convergence Science and Technology, Asan Medical Center, University of Ulsan College of Medicine, Seoul, Korea, 5 Department of Convergence Medicine, University of Ulsan College of Medicine, Seoul, Korea

☯ These authors contributed equally to this work.
\* heuiran@amc.seoul.kr (HL); gsjeong@amc.seoul.kr (GSJ)

**Data Availability Statement:** All relevant data are within the manuscript.

## Abstract

As a new class of cancer therapeutic agents, oncolytic viruses (OVs) have gained much attention not only due to their ability to selectively replicate in and lyse tumor cells, but also for their potential to stimulate antitumor immune responses. As a result, there is an increasing need for *in vitro* modeling systems capable of recapitulating the 3D physiological tumor microenvironment. Here, we investigated the potential of our recently developed microphysiological system (MPS), featuring a vessel-like channel to reflect the *in vivo* tumor microenvironment and serving as culture spaces for 3D multicellular tumor spheroids (MCTSs). The MCTSs consist of cancer A549 cells, stromal MRC5 cells, endothelial HUVECs, as well as the extracellular matrix. 3D MCTSs residing in the MPS were infected with oncolytic VSV expressing GFP (oVSV-GFP). Post-infection, GFP signal intensity increased only in A549 cells of the MPS. On the other hand, HUVECs were susceptible to virus infection under 2D culture and IFN-β secretion was quite delayed in HUVECs. These results thus demonstrate that OV antitumoral characteristics can be readily monitored in the MPS and that its behavior therein somewhat differs compared to its activity in 2D system. In conclusion, we present the first application of the MPS, an *in vitro* model that was developed to better reflect *in vivo* conditions. Its various advantages suggest the 3D MCTS-integrated MPS can serve as a first line monitoring system to validate oncolytic virus efficacy.

## Introduction

Oncolytic viruses (OVs) selectively infect and destroy tumor cells, sparing the normal cells and minimizing normal tissue damage [1]. These characteristics of OVs have enabled the growth

**Funding:** This work was supported by grants from the Basic Science Research Program through the National Research Foundation of Korea (NRF, http://www.nrf.re.kr) granted funded by the Korea government (NRF-2016R1A2B4014912 to CH Joo; NRF-2019R1A2C2005244 to GS Jeong; NRF-2019R1C1C1007468 to KJ Lee). The funder had no role in study design, data collection and analysis, decision to publish, or preparation of the manuscript.

**Competing interests:** The authors declare no competing interests.

of oncolytic virotherapy combined with immunotherapy [2]; consequently, oncolytic virotherapy has become one of the burgeoning fields of cancer immunotherapy [3]. To improve antitumor therapeutic efficacy, native features of OVs have been optimized by genetically modifying the viruses in various ways by inserting immune-stimulating genes and by removing cytotoxic viral genes [4]. For example, Talimogene Laherparepvec (T-VEC) acquired the first FDA approval as an oncolytic virus therapy with the recent completion of phase III clinical trials.

Two dimensional (2D) *in vitro* cell culture systems are routinely applied to determine the characteristics of OVs [5]. However, 2D systems do not reflect the 3D physiological microenvironments where the tumors reside [6]. Tumor xenograft models including mouse [7] and rat [8] are widely used to evaluate the tumor killing efficacy of OVs, but are still not able to accurately predict the outcome of human clinical trials [9]. OVs have dual therapeutic effects that depend on onco-selective cell lysis and the induction of antitumor immune responses [10]. Yet, *in vivo* tumor xenograft models do not show induced immune responses following OV administration. Therefore, well-defined *in vitro* models that imitate the *in vivo* cancer microenvironment conditions would be more beneficial as a first line study platform. Among several types of 3D-cultured spheroid models [11], 3D multicellular tumor spheroids (3D MCTSs) show suitable *in vivo* environments for evaluating the properties of onco-selective infection of OVs [12]. Another approach is applying a microphysiological system (MPS) to simulate blood vessel-like structures [13,14]. If the MPS and 3D MCTSs are properly combined, it would serve as a much better *in vitro* tumor model for evaluating the efficacy of OVs because it mimics the actual physiological conditions of the tumor tissue, including the fluid dynamics and cell-to-cell interactions in the TME.

Recently, we successfully invented a model MPS combined with 3D MCTSs and demonstrated that the microfluidic device could be adopted for the development and screening of anticancer chemical drugs by the formation of the same 3D MCTSs under similar conditions [15]. Here, we report a novel approach to investigate the antitumor activities of oncolytic vesicular stomatitis virus (oVSV) through employing our newly developed MPS. By employing oVSV-expressing GFP (oVSV-GFP), the data provide evidence that the antitumoral characteristics of oVSV can be readily monitored in the 3D MCTS-integrated MPS and that its behavior somewhat differs in this condition compared to that in a conventional 2D system. The 3D MCTS-integrated MPS thus can serve as the first line evaluation system for the onco-selective infection of OVs.

## Materials and methods

### Cell cultures, fluorescence labeling, and fluorescence analysis

Human lung cancer A549 cells (ATCC, Manassas, VA) and human lung fibroblast MRC5 (ATCC) were cultured in DMEM (Thermo Fisher Scientific, Waltham, MA) containing 10% FBS (Biowest, Riverside, MO) and 1% penicillin/streptomycin (Thermo Fisher Scientific). HUVECs were maintained in endothelial cell medium (Sciencell Research Laboratories, Carlsbad, CA).

According to the manufacturer's protocol, cell components were fluorescently labeled in 3D MCTSs using green PKH67GL or red PKH26 (Sigma-Aldrich; ST. Louis, MO). Briefly, the cell pellet was suspended in Diluent C solution, and the dye solution was prepared with a PKH ethanolic dye solution. The suspended cells and the prepared dye solution were rapidly mixed. After incubation, 1% bovine serum albumin (BSA) was added to stop the staining. After centrifugation, the cells were resuspended in complete medium, centrifuged and washed to ensure the removal of any unbound dye. After washing, the fluorescent dye-stained cells were used in experiments to confirm their position and migration.

Fluorescence images were developed using a fluorescence microscope (EVOS) and confocal microscope (Zeiss LSM780, Carl Zeiss AG, Oberkochen, Germany). The confocal images were analyzed using ZEN Microscope Software (Carl Zeiss AG, Oberkochen, Germany).

## Oncolytic VSV expressing GFP (oVSV-GFP) preparation and single growth curve analysis

The VSV full-length plasmid with GFP tagging, pVSV Venus-VSVG (Addgene, Watertown, MA), was the template to generate oVSV-GFP. It was generated as previously described [16]. Briefly, 293T cells were infected with vaccinia virus encoding T7 polymerase (ATCC, vTF7-3) at 10 MOI. After 1 h, the vaccinia virus was washed, and the cells were transfected with 1.1 μg of pVSV Venus-VSVG, 0.6 μg of pN, 1.4 μg of pP, and 0.9 μg of pL by using 32 μl of PEImax (Polyscience; Germany). After 48 h, filtered culture media were treated with new 293T cells. oVSV-GFP present in the supernatant were further passaged in 293T cells to amplify the viral titer. A549 cells were used to quantify the infectious titer of oVSV-GFP virus stock by $TCID_{50}$. Cells treated with oVSV-GFP for 1 h, at 1 multiplicity of infection (MOI) representing one infectious virus per cell, were used in single growth curve analysis. Fresh growth media was added after washing and the supernatants were collected at designated time points. Virus titers were also determined using the supernatants by $TCID_{50}$.

## Generation of the microphysiological system (MPS)

**i) Fabrication of the microfluidic culture device and passive powerless micropump.** The manufacturing procedure of the polydimethylsiloxane (PDMS; Sylgard® 184, Dow Chemical Co., MI)-based microfluidic device was described in our previous study [15]. A passive micropump with a siphon effect was established as well. This pump provided continuous medium without mechanical energy. A polyethylene (PE) tube was linked between the device and the micropump. Eluate was drained into the outlet reservoir via a PE tube filled with yarn fiber. After connecting the pump to the microfluidic device, the external structure of the MPS was completed.

**ii) Generation of the tumor microenvironment (TME)-like structure with 3D MCTSs.** The generation of 3D MCTS with ECM in a microfluidic device was established in our previous studies [17]. The internal device was coated with type I collagen solution (collagen I; BD Bioscience, San Jose, CA) to simulate an ECM. A549 and MRC5 cells (2 x $10^6$ cells/mL) were suspended in 2 mg/mL type I collagen solution. Each cell line was injected into the microfluidic device and centrifuged to trap the cancer cells in the wells. Afterwards, the microfluidic device was washed with a collagen solution. HUVECs (1 x $10^6$ cells/mL) were then seeded in a channel to generate a vessel-like environment. The number of cells remaining per chip was identified as an average of 1 x $10^5$ cells of each line. The microfluidic device was connected to the passive micropump to supply continuous medium, so the internal components of the microphysiological system (MPS) were completed. For LDLR expression studies, MPS was prepared only using HUVEC cells while all other procedures remained the same.

## oVSV-GFP infection of the MPS and virus growth kinetics

At 24 h after preparing MPS, a microtip containing oVSV-GFP at 5 MOI based on the total number of cells per MPS (3 x $10^5$ cells) was inserted into the inlet hole of the MPS. The height difference between the microtip containing fluid and chip was used to introduce viruses into MPS channels via pressure and flow. After 1 h, residual viruses were removed, and the infected MPS was washed. The MPS was then connected with the passive micropump to supply oxygen

and nutrients. The drained eluate from the MPSs were collected at 24, 48 and 72 h to measure infectious viral particles (IPs) by $TCID_{50}$.

## Cell death assay

For propidium iodide (PI) analysis, cells were stained with PI and DAPI solution in a 2D culture and examined under a fluorescence microscope at ×200 magnification. To evaluate antitumor effects by oVSV-GFP in MPS, cellular mortality was analyzed by fluorescence staining using a live/dead kit according to the manufacturer's protocol. Morphologies and fluorescence images of 3D MCTSs in the MPS were examined using a fluorescence microscope at 24, 48 and 72 h after oVSV-GFP infection.

## RT-quantitative PCR (qPCR)

QIAamp viral RNA mini kits (QIAGEN) were used to recover viral RNA genome from eluates. Reverse transcription was performed using SuperScript III Reverse Transcriptase (Life Technologies). qPCR was performed using IQ SYBR Green Supermix (Bio-Rad; Hercules, CA) in a real-time thermal cycler (CFX96 Real-time system; Bio-Rad). The following sequences were selected: VSV-G, 5'-CCCGGTACCTTTTTCTTTATCATAGG GT -3', 5'-CCCGTTAACTT ACTTTCCAAGTCGGTT-3'.

## Susceptibility analysis of cell lines against oVSV infection in a 2D culture system

Cells were plated and infected with serially diluted oVSV-GFP for 1 h at 37˚C. After 5–6 days, the cells were stained with 2% crystal violet in 50% methanol. The lowest MOI of each cell line for cytotoxicity was calculated by $TCID_{50}$. Based on the lowest MOI of A549, the relative susceptibility of each cell line against oVSV-GFP was calculated (n = 4).

## Quantification of secreted human interferon-beta (IFN-β)

Cells in 2D culture were infected with oVSV-GFP at 1 MOI for 1 h, then washed and given fresh media. Culture media was collected at 6 h and 14 h, then stored at -80˚C prior to analysis. The concentration of IFN-β was measured using a human IFN-β ELISA kit (PBL Assay Science, Piscataway, NJ).

## Immunofluorescence staining for detecting the expression of LDLR

After fixing with 4% paraformaldehyde (Sigma-Aldrich) and subsequently washing with PBS, 3D MCTSs were incubated with permeabilizing solution using PBS containing 0.1% Triton X-100 (Sigma-Aldrich) and blocked with 2% BSA (Sigma-Aldrich) in 0.1% Tween 20 (Sigma-Aldrich). The 3D MCTSs were incubated with a fluorophore-conjugated antibody against low-density lipoprotein receptor (LDLR; Bioss, Woburn, MA) in 1% BSA in PBS overnight at 4˚C, washed with PBS and stained with DAPI (Invitrogen).

## Statistical analysis

The area and fluorescence intensity of the MCTSs were determined by ImageJ software (1.46 ver, NIH). Quantitative data are presented as the mean ± standard deviation (SD). Group differences were assessed by paired t-tests or one-way and two-way ANOVAs followed by Bonferroni's test using GraphPad Instat (GraphPad Software, La Jolla, CA). Statistical significance was set at $p > 0.05$(ns), $p < 0.05$(*), $p < 0.01$(**) and $p < 0.001$(***).

## Results

### *In vitro* microphysiological system harboring 3D MCTS for the evaluation of oVSV-GFP activity

To successfully establish an appropriate *in vitro* tumor model reflecting the *in vivo* TME, we employed our recently developed MPS [15] (Fig 1). Oxygen, nutrients, and components simulating the TME were constantly provided through a vessel-like channel in the integrated MPS using a passive micropump (Fig 1A). The internal components forming the TME-like environment in the MPS were ECM (type I collagen matrix) and 29 3D MCTSs located in 29 wells (Fig 1B). In particular, 3D MCTS consisting of human cancer A549 cells (green, Fig 1C) and human fibroblast MRC5 cells mimic the structure of *in vivo* tumor tissues. (red, Fig 1C). As expected, cells observed in the 3D overlap region of MCTS appeared yellow in the merged image. The fluorescence image suggests the successful lining of HUVEC cells in the MPS (Fig 1D, red).

### Effective replication of oVSV-GFP in A549 cells

To evaluate the efficacy of oncolytic virotherapy, we employed oVSV encoding GFP (Fig 2A). By one-step growth curve analysis, A549 human cancer cell susceptibility to oVSV-GFP was determined (Fig 2B). The viruses rapidly replicated, reaching a plateau at approximately 25 h p.i. ($6.6\pm4.2\times10^7$ $TCID_{50}$/ml). Unlike untreated cells (Fig 2C), infected cells showed irreversible cytopathic effects and strong GFP signaling under fluorescence microscopy (Fig 2D). The PI staining results indicated that A549 cancer cells are highly susceptible to oVSV-GFP replication, resulting in cancer cell death and subsequent progeny virus release.

### Differential replication of oVSV-GFP in the 3D MCTS-integrated MPS

Viral infection with oVSV-GFP at 5 MOI (determination based on the total number of cells in MPS) was achieved through natural flow and subsequent MPS washing (Fig 3A). GFP signaling in the fluorescence images of 3D MCTSs was evident at 24 h p.i., then gradually declined (Fig 3B and 3C). Using the eluates released from the MPS every 24 h, progeny virus production was quantified by $TCID_{50}$ (Fig 3D) and VSV-G gene (Fig 3E) specific RNA genome amplification was measured by RT-qPCR. Similar to the change in fluorescence intensity, progeny virus production was the highest at 24 h ($1.1\pm0.4\times10^6$), and then decreased at 48 h ($3.2\pm1.3\times10^4$) and 72 h ($4.8\pm3.1\times10^4$). The peaking of production at 24 h followed by decline was also observed in RNA genome analysis of the virus.

The replication ability of oVSV-GFP in different cells consisting of the 3D MCTSs in the MPS was examined by labeling A549 and MRC5 cells with a red fluorescence tracker. At every time point p.i., GFP signals were clearly observed in red A549 cells. This is indicated in the yellow merged images (Fig 4A). In contrast, red MRC5 cells did not turn yellow following oVSV-GFP replication (Fig 4B). Though the tumor spheroids consisted of both cells, the data indicates that GFP signal following oVSV replication was restricted in A549 cells. To further investigate the characteristics of the cells in MCTSs after oVSV-GFP treatment, the tumor spheroids unlabeled with fluorescent dye were analyzed with PI staining (Fig 4C). The PI-positive dead cell region perfectly overlaps with the GFP-positive region. This indicated that the GFP-positive A549 cells, but not the MRC5 cells or HUVECs, were dead. Finally, we observed that the size of infected MCTSs gradually decreased when that of uninfected control MCTSs increased (Fig 4D and 4E). Taken together, these results demonstrate that oVSV-GFP selectively replicates in A549 human cancer cells and eventually induces cell death in infected cells in the MPS.

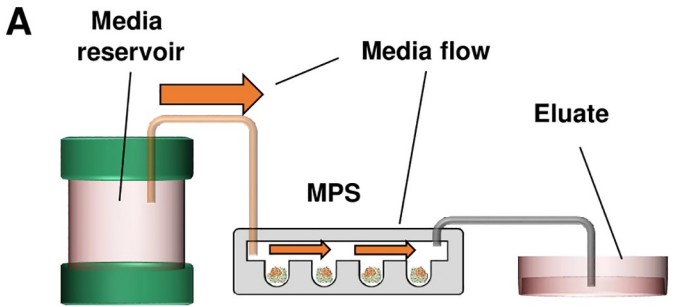

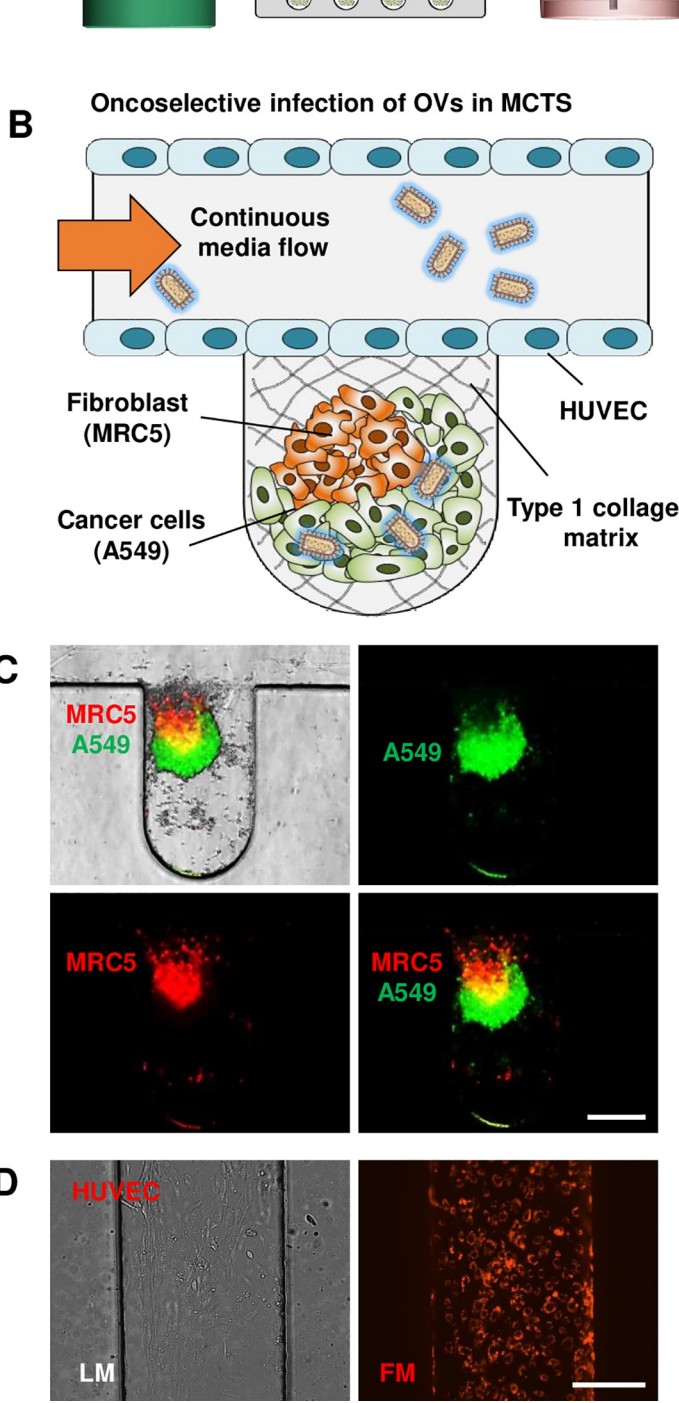

**Fig 1. The structure of the microphysiological system (MPS).** (A) Schematic representation of the external structure of the MPS. (B) Schematic of the internal components of the MPS. In the well, collagen type I matrix was coated as ECM, and 3D MCTSs were generated with human cancer A549 cells and human fibroblast MRC5 cells. In the vessel-like channel, HUVECs were coated, and continuous medium was provided to the 3D MCTSs. (C) Fluorescent images of a labeled 3D MCTS in the MPS (A549, green; MRC5, red). (D) Images of HUVECs stained with red in the MPS; scale bars: 275 μm.

## Different susceptibilities of endothelial cells to oVSV in the MPS environment

To examine whether the physiological environment of the MPS influences the susceptibility of the 3 different cells residing in the MPS, each cell line was infected with oVSV-GFP at 1 MOI

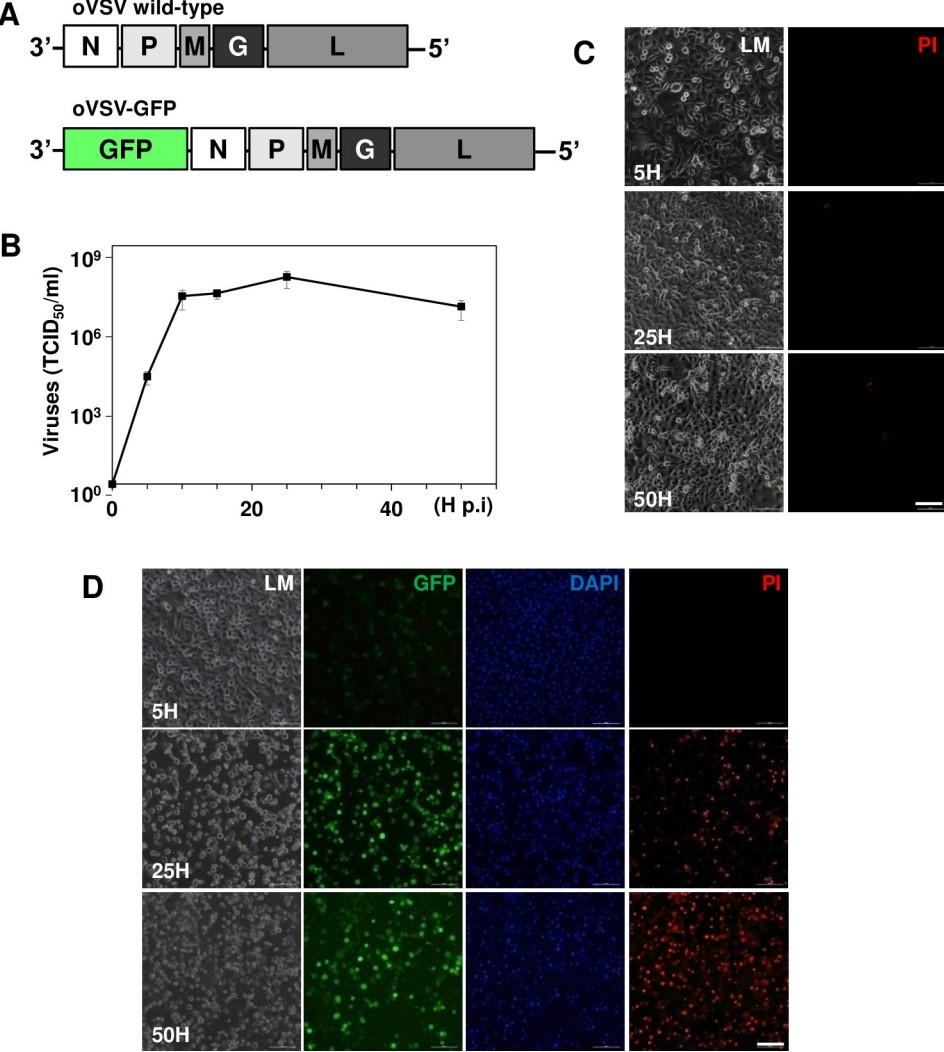

**Fig 2. Growth properties and cytopathic effects of oVSV-GFP.** (A) Schematic drawing of oVSV-GFP. GFP was inserted in front of gene N of viral genome. (B) Single growth curve analysis of oVSV-GFP in A549 cells. The supernatants were collected from infected A549 cells at the indicated time points (0, 5, 10, 15, 25h and 50 h p.i.), and the virus titration was calculated by $TCID_{50}$ assay. (C) The characteristics of mock-treated cells at specified times. (D) Cytopathic effects of oVSV-GFP in infected cells. GFP-positive cells were identified starting at 5 h p. i., and PI-positive cells were easily identified at 25 h and 50 h. The mean and SE data of 3 independent experiments; scale bars: 200 μm.

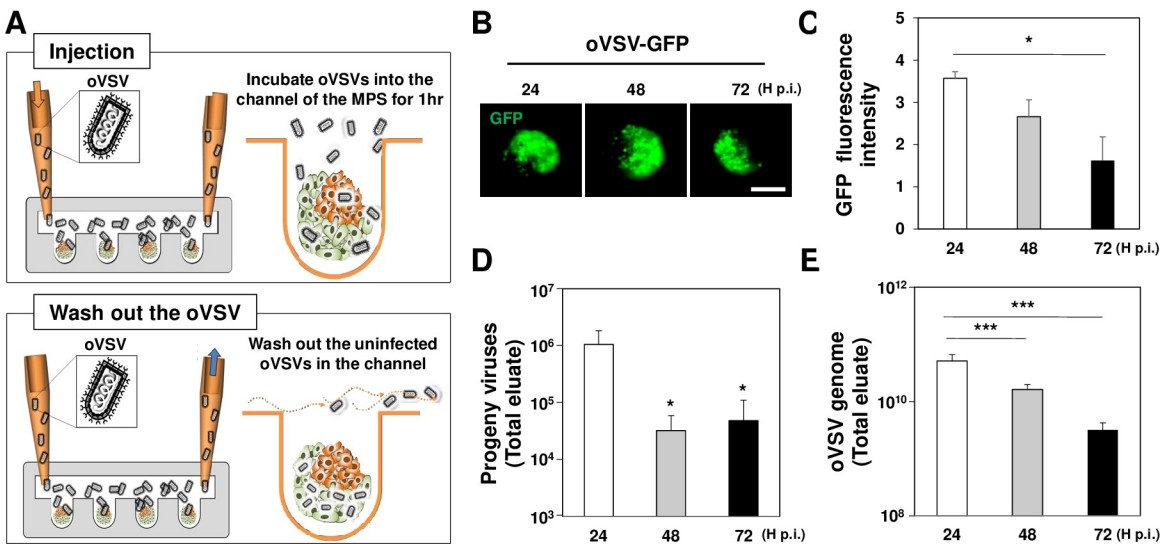

**Fig 3. Effective replication of oVSV-GFP in the MPS.** (A) Schematic representation of virus infection. As described in M&M, viruses at 5 MOI were introduced into MPS. The residual viruses were removed by washing, and the infected MPS was connected to a passive micropump. (B) Fluorescence images of a 3D MCTS infected by oVSV-GFP. (C) At the indicated time points, a fluorescence intensity graph of oVSV-GFP was obtained from 3D MCTS (tumor spheroid; n = 8). (D) Quantitative analysis (n = 3) of progeny virus production by $TCID_{50}$ assay using the eluates at the indicated time points. (E) Viral RNA genome copies from the eluates were calculated by RT-qPCR targeting VSV-G gene (n = 3). In addition to the results in C and D, the viral genome of oVSV-GFP at 24 h was the highest. The mean and SE data of 3 independent experiments; scale bars: 200 μm. $^{*}p < 0.05$, $^{***}p < 0.001$.

in a conventional 2D culture system, and the relative sensitivity to oVSV was analyzed (Fig 5A). In 2D, A549 cells were highly susceptible to oVSV while MRC5 cells were relatively resistant, similar to the MPS model. However, HUVECs were very sensitive to oVSV in 2D only. The influence of IFN-β was investigated by quantifying IFN-β secretion from the culture medium of each cell line following oVSV-GFP treatment in 2D culture using ELISA (Fig 5B). A549 cells constantly secreted IFN-β at very low levels, while MRC5 cells rapidly secreted IFN-β at high concentrations, reflecting the susceptibility to oVSV. In the case of HUVECs, IFN-β secretion showed a delayed pattern; in the early phase of infection (6 h p.i), its level was very low, which was even lower than that for A549 cells. These results indicate that the delayed IFN-β secretion in the infected HUVECs could have a correlation with their susceptibility to oVSV in the 2D system. The effect of fluid flow on the expression of low density lipoprotein receptor (LDLR), the main receptor of oVSV, was further investigated in HUVECs after virus treatment (Fig 5C). In MPS consisting of HUVECs alone, substantial GFP signals were observed regardless of fluidic flow compared to MPS with all three cell lines (Fig 4). Similarly, LDLR expression was also detected. Thus, the data suggest the susceptibility of HUVEC cells in MPS without tumor spheroid and the consistent LDLR expression levels regardless of fluidic flow.

## Discussion

To establish an advanced *in vitro* tumor model reflecting the *in vivo* TME for better evaluating the antitumor efficacy and safety of oncolytic viruses, we employed our recently developed MPS and examined its potential [15]. A durable 3D structure was formed with HUVEC lining, 3D MCTS composed of cancer A549 cells, and fibroblast MRC5 cells. When connected to a passive micropump, the MPS channel supports continuous media supply. oVSV expressing GFP can be readily applied to this 3D MCTS, and the virus replication pattern can be

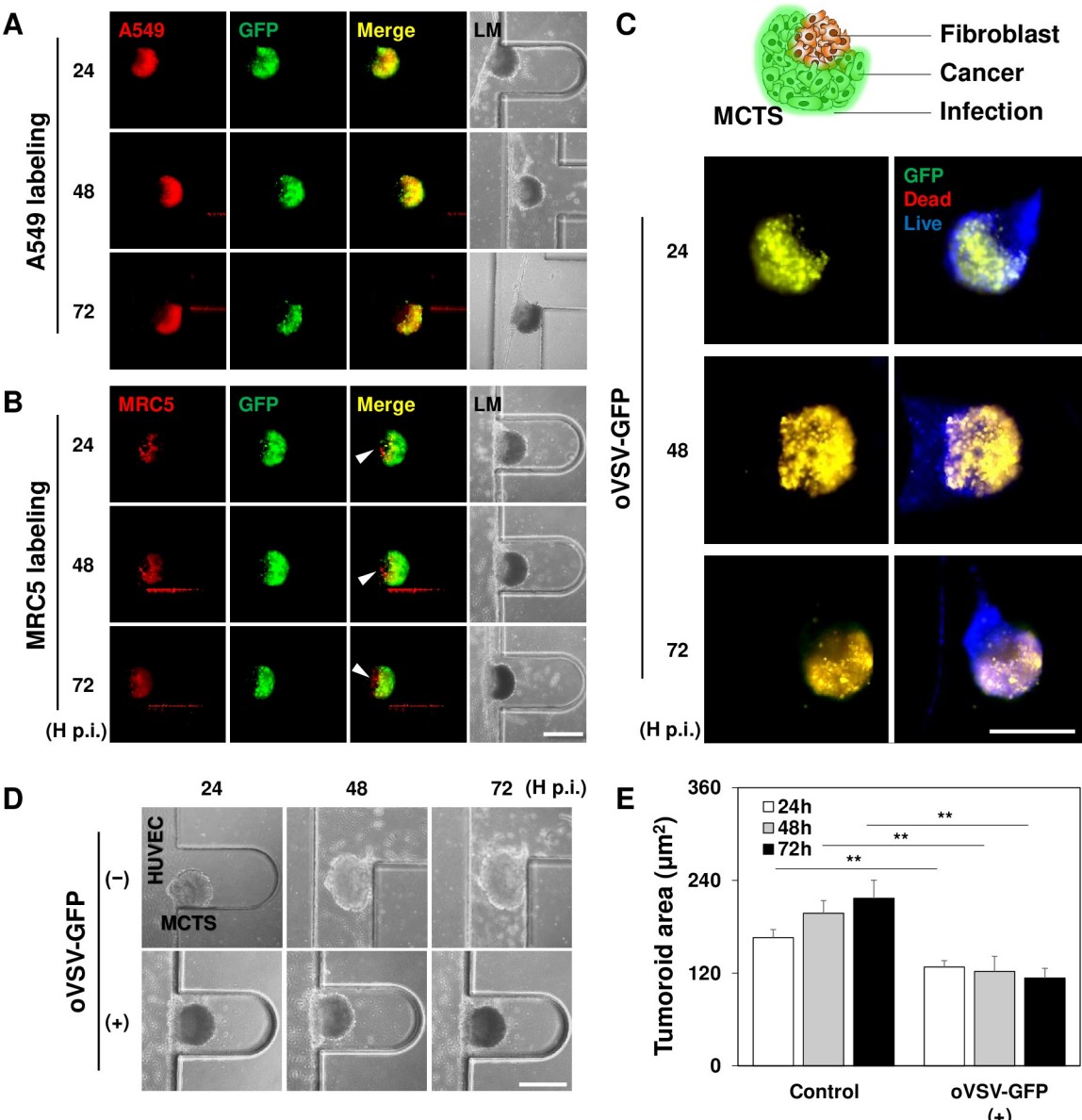

**Fig 4. oVSV-GFP selectively kills A549 cells following virus replication in the MPS.** (A) Infection patterns of the MPS generated by A549 cells marked with a red fluorescence tracker at the indicated time points (tumor spheroid; n = 8). The locations of the GFP signal are similar to those of labeled A549 cells. (B) Infection patterns of the MPS generated by MRC5 cells marked with a red fluorescence tracker at the indicated time points (tumor spheroid; n = 8). The locations of the GFP signal did not overlap with those of labeled MRC5 cells (indicated with white arrows). (C) PI staining with 3D MCTSs infected with oVSV-GFP at the designated time points (tumor spheroid; n = 8). The locations of the GFP signal perfectly overlaped with those of dead-staining cells. (D) Light microscopy images of 3D MCTSs infected with oVSV-GFP (tumor spheroid; n = 10). (E) Growth kinetics of the MCTSs at the indicated time points (tumor spheroid; n = 10). Scale bars: 400 μm. ** $p < 0.01$.

visualized under fluorescence microscopy in real time. The present study indicates that oVSV preferentially infects cancer cells and that the morphology of the 3D MCTS collapses as the infected cells die. Based on these advantages, as an *in vitro* tumor model, the MPS offers a convenient monitoring method for validating the efficacy of oncolytic viruses.

VSV is considered a particularly promising OV due to its effective antitumor activity in preclinical models and has the following advantages [18]: i) it has a natural tropism for malignant

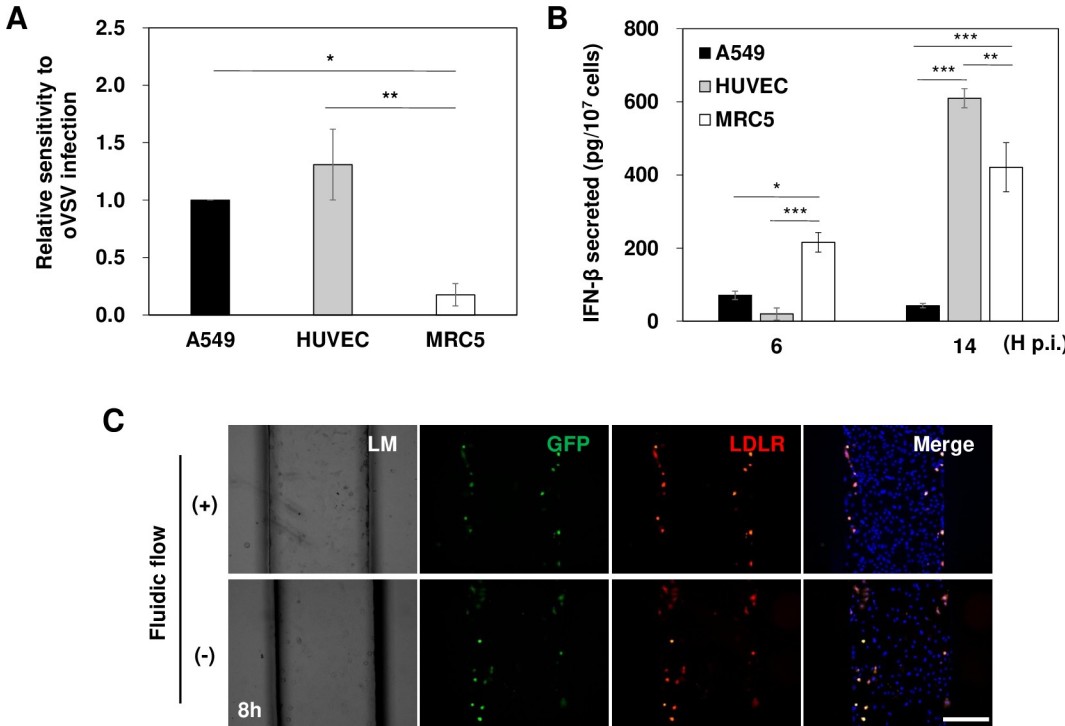

**Fig 5. The fluidic flow in the MPS influences the susceptibility of HUVECs to oVSV.** (A) Relative susceptibility of three cell lines to oVSV without continuous fluidic flow conditions 2D culture system. (B) Comparison of IFN-β secretion in the culture medium of each cell line after oVSV-GFP treatment in 2D culture. (C) LDLR expression and GFP signal monitoring after 8 h oVSV-GFP infection in the MPS consisting of HUVEC cells alone with or without fluidic flow. The mean and SE data of 3 independent experiments; scale bars: 275 μm. $^{*}$p < 0.05, $^{***}$ p < 0.001.

cells because of its high sensitivity to type I IFN, thus protecting normal cells from destruction [19]; ii) virtually no preexisting immunity against VSV exists in the general population, and the rare natural infections are asymptomatic; iii) it has a rapid replication cycle and high titers and can thus rapidly destroy cancer cells; and iv) it induces substantial immunogenicity to stimulate antitumor immunity. Recently, the Russell group developed a genetically modified oVSV expressing IFN-β and sodium iodide symporter and examined this oVSV in phase I clinical trials for patients with refractory multiple myeloma, acute myeloid leukemia, or T-cell lymphoma [20].

Unlike A549 or MRC cells, HUVEC cells showed a difference in viral sensitivity in 2D and 3D culture conditions. HUVEC cells, which were sensitive to oVSV-GFP in the 2D culture system, were resistant in the 3D MPS. Since the viral sensitivity is presumed to be due to the difference in IFN signaling in each infected cell line [21,22], IFN-β secretion was measured by ELISA. The IFN-β secretion pattern correlated with the oVSV sensitivity of each cell line in 2D. Uninfected cells can be efficiently protected from secondary infections through the paracrine signaling of IFN secreted by nearby infected cells [23,24]. Previous *in vivo* studies also showed that tumor cells are infected and killed by VSV while normal endothelial cells are protected [25,26]. Therefore, the low susceptibility of HUVECs to oVSV in the MPS may be explained by the protective effect of IFN secretion by neighboring cells against viral infection.

LDLR is a major surface receptor of mammalian cells for VSV infection through endocytosis [27], and LDLR activity plays an important role in determining the degree of infection of VSV [28]. The present study clearly demonstrates consistent LDLR expression levels in

HUVEC cells regardless of fluidic flow and may suggest that differences in HUVEC sensitivity between MPS and 2D systems are not related to LDLR expression on the cell surface. The effect of fluidic flow on LDLR activity has been also illustrated in previous studies showing a decrease of LDLR activity in vascular endothelial cells *in vitro* in the presence of microscopic or low degree laminar shear stress (0.1 ~ 0.5 dyne/$cm^2$) [29]. To identify the shear stress that affects LDLR activity, we previously demonstrated that laminar flow occurs in the microchannels of the microfluidic, with the resulting laminar shear stress in a microchannel to be 0.2 ~ 0.3 dyne/$cm^2$ [17]. It is thus possible to conclude that the laminar flow in our microfluidic system, which reflects the *in vivo* TME, affects HUVEC LDLR activity in the microchannel which results in low LDLR activity. Further research is still needed to elucidate the effect of shear stress on the viral sensitivity of endothelial cells in detail.

To apply the 3D microfluidic MPS *in vitro* and assess the therapeutic potential of oncolytic virotherapy, quantitative analysis is essential. In our MPS, the virus replication pattern can be monitored by tracing the fluorescence signal in real time. Noninvasive repeated monitoring of the MCTSs residing in each well is possible under conventional fluorescence microscopy. The fluorescent area can be quantified using proper software, and then the data can be further analyzed for evaluating the oncolytic efficacy of OVs. Analysis via immunocytochemistry is also possible. Additional quantitative data can be obtained utilizing eluate released from MPS, including viral growth kinetics or total viral genomic quantification, where $TCID_{50}$ or RT-qPCR is performed. Furthermore, the 3D MCTSs residing in the MPS can be used to analyze total protein or RNA expression patterns upon extracting the MCTS pellets from the device.

Although this 3D MCTS-integrated MPS has several valuable advantages for evaluating the potential of oncolytic virotherapy, some limitations exist, such as a lack of immune response-related cells. Immune cells play a key in switching from hot to cold tumor conditions, attenuating cold tumor properties driving the interactions among tumor cells, other resident cells, and oncolytic viruses, particularly when cancer destruction occurs [30,31]. Since the direct killing of tumor cells by OVs and subsequent killing by activated immune cells are important points for cancer removal [32], the lack of immune cells in the system limits the current study. Subsequent studies can explore how to apply immune cells to this system and its effects in evaluating oncolytic virotherapy.

In conclusion, we present the first application of an MPS-harboring 3D MCTS as an *in vitro* model for the validation of the antitumor effects of an oVSV. The 3D MCTS-integrated MPS was developed to better reflect the *in vivo* environment and is an improved *in vitro* tumor model system compared to the 3D MCTS system because it simultaneously incorporates 3D MCTSs and a microfluidic device. Due to these advantages, the 3D MCTS-integrated MPS can serve as the first line monitoring system for successfully validating the efficacies of OVs, which may prove to be a valuable model system for future evaluations of various anticancer therapies beyond OVs.

## Acknowledgments

We thank Paula Khim for her important advice on editing a draft of this manuscript.

## Author Contributions

**Conceptualization:** Gi Seok Jeong, Heuiran Lee.

**Investigation:** Kyoung Jin Lee, Sang Woo Lee, Ha-Na Woo, Chul Hyun Joo, Gi Seok Jeong, Heuiran Lee.

**Methodology:** Kyoung Jin Lee, Sang Woo Lee, Hae Mi Cho, Dae Bong Yu, Soo Yeon Jeong, Heuiran Lee.

**Supervision:** Gi Seok Jeong, Heuiran Lee.

**Writing – original draft:** Kyoung Jin Lee, Heuiran Lee.

**Writing – review & editing:** Kyoung Jin Lee, Sang Woo Lee, Ha-Na Woo, Chul Hyun Joo, Gi Seok Jeong, Heuiran Lee.

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
