## [Decision Letter · Decision Letter 0]

6 Mar 2020

PONE-D-19-34362

Real-time monitoring of oncolytic VSV properties in a novel in vitro microphysiological system containing 3D multicellular tumor spheroids

PLOS ONE

Dear Dr. Lee,

Thank you for submitting your manuscript to PLOS ONE. After careful consideration, we feel that it has merit but does not fully meet PLOS ONE’s publication criteria as it currently stands. Therefore, we invite you to submit a revised version of the manuscript that addresses the points raised during the review process.

As you can see, both reviewers recommended significant changes to your manuscript.  While no new experiments may be needed to address the critiques, both reviewers have asked for a number of clarifications.

We would appreciate receiving your revised manuscript by Apr 20 2020 11:59PM. To enhance the reproducibility of your results, we recommend that if applicable you deposit your laboratory protocols in protocols.io, where a protocol can be assigned its own identifier (DOI) such that it can be cited independently in the future. For instructions see: http://journals.plos.org/plosone/s/submission-guidelines#loc-laboratory-protocols

We look forward to receiving your revised manuscript.

Kind regards,

Salvatore V Pizzo

Academic Editor

PLOS ONE

Journal Requirements:

Reviewers' comments:

Reviewer's Responses to Questions

**Comments to the Author**

1. Is the manuscript technically sound, and do the data support the conclusions?

Reviewer #1: No

Reviewer #2: Partly

2. Has the statistical analysis been performed appropriately and rigorously? 

Reviewer #1: No

Reviewer #2: Yes

3. Have the authors made all data underlying the findings in their manuscript fully available?

Reviewer #1: Yes

Reviewer #2: Yes

4. Is the manuscript presented in an intelligible fashion and written in standard English?

Reviewer #1: Yes

Reviewer #2: Yes

5. Review Comments to the Author

Reviewer #1: The authors previously developed a microphysiological system (MPS) which features a vessel-like channel that can be lined with endothelial cells and a total of 29 side chambers which can be populated with various cell types. In the current study they generated spheroids in the side channels by infusing A549 cancer cells and MRC5 fibroblasts, then lined the vessel-like channels with HUVECs. They infused VSV-GFP into the vessel-like channels and were able to show that the virus infected only the A549 cells, not the MRC5 cells in the spheroids. This result was expected since the MRC5 cells were resistant to the virus in 2D culture and the A549 cells are susceptible. Interestingly the HUVECs did not get infected in the MPS, but were susceptible to virus infection in 2D culture. The authors speculate based their analysis of IFNb release from VSV infected A549, MRC5 and HUVEC cells, and the protection of those cells from VSV infection upon IFNb exposure, that the release of IFNb by virus-exposed MRC5 cells in the MPS is able to render the HUVEC resistant to infection.

The MPS system colonized with multicellular spheroids does offer an interesting in vitro test system for the study of oncolytic viruses that might well catch on as an alternative to mouse xenograft models if it were widely available. The ability to form multicellular tumor spheroids in side chambers accessible only via a vessel-like channel is appealing and allows for detailed analyses of the dynamical aspects of virus interaction with microvessels, tumor cells and fibroblasts in a controllable system. However, there are significant weaknesses with the current manuscript which, in the opinion of this reviewer, should be addressed prior to publication:

1. The authors state that they perfuse their system with VSV-GFP using a virus moi (multiplicity of infection) of 5.0. But what does that mean? First, they do not give details of the cell substrate on which they titrate their virus stock. Second, it is not clear how they determine the "moi" for the MPS when they do not know how many cells are present in the MPS that they are perfusing. This aspect of the paper is confusing and needs to be clarified.

2. The vessel-like channels in the MPS are supposedly lined by HUVEC cells but there are no images provided to confirm this, nor to show whether the HUVEC lining is indeed sufficiently comprehensive to form vessel-like channels. This is a critical aspect of the system and should be addressed with additional data. Without a direct visualization of the HUVEC lining cells, and an assessment of their GFP expression post virus infusion, the claim that the HUVEC cells resident in the MPS are resistant to virus infection remains unproven.

3. The authors state that there are 29 side chambers in their MPS but do not indicate how they captured data from all of these chambers and do not provide any statistical backup for the conclusions they have drawn.

4. The hypothesis that HUVEC cells lining the MPS channels are rendered resistant to VSV-GFP by exposure to IFNb from MRC5 cells could easily be tested by rerunning the experiment using A549-only spheroids that do not contain MRC5 cells . Data from such an experiment should be included in the manuscript.

Reviewer #2: Authors have investigated the potential of their recently developed microphysiological system (MPS) that mimics the in vivo tumor microenvironment and serves as culture space for 3D multicellular tumor spheroids (MCTS). This system was used as a model to check the oncolytic potential of Vesicular Stomatitis virus and concluded that this system can be used as a first line of monitoring system to validate any oncolytic virus efficacy. It is a helpful model system for evaluation of anticancer therapies including virotherapy.

It is a good piece of work; however it was not presented very well. Methodology and results need to be written clearly. At many places the clarity is missing. Experiment description in the legends needs lot of improvement.

Comment 1: In Figure 1C, A549 cells stained green and MRC5 red. But why is it bright yellow at the interface of these two cell populations? Is it due to overlap?

Comment 2: Reference to the “effective replication of oncolytic VSV expressing GFP in A549cells” under result section, related Fig 2C, oVSV effects were shown at 5h and 50h. It would be better, if 25h data of GFP expression and PI staining is also incorporated in the image panel. In this experiment, CPE/GFP expression/cell death was measured up to 50h post infection; but the negative control (mock infected) was not included. How was it confirmed without a negative control that the noted cell death was virus induced?

Comment 3: In Figure 3E, quantification of viral genome copies was done to analyse oVSV-GFP replication in MCTS eluates. How was it done? Was it the amplification of VSV –G gene by quantitative PCR? Please clarify in the text and legend. In this experiment as well, negative control was missing.

Comment4: What was the rationale behind using different MOIs, 1 and 5 for infections in 2D cultures and MPS?

Comment 5: From the methodology section given under ‘quantification of secreted human interferon beta (IFNβ) after VSV-GFP infection’ it is clear that IFNβ response was measured in all three types of cells following oVSV-GFP infection in 2D cultures and eluates collected from infected MPS by ELISA. But the related result and Figure/ figure legend are confusing. Please check the result and the figure related to this experiment and write more in detail.

Comment 6: To check the differential replication of oVSV-GFP in the 3D MCTS integrated MPS, A549cells, MRC-5 cells were labeled with fluorescence tracker. Figure 4A and 4B, both the cells; A549 and MRC5 were stained red. How were these two cell populations separated to check the expression oVSV-GFP through red and green merge? Are 4A and 4B images from the same spheroid? Clarify and also mention in the text and legends. In the figure 1, A549 cells were stained green.

Comment 7: There is a lot of room for improvement in discussion section. One sentence in the discussion from lines 326 to 329 is the exact repetition of a sentence under result section from lines 281 to 284. Improve the discussion.

Comment 8: The intensity of GFP expression in A549cells does indicate the active replication of oVSV-GFP virus but not the anti tumor activity. Experiment of Live/Dead cell staining following virus infection does not commend the anti tumor activity of oncolytic virus used. An anti tumor/ anti proliferative marker may be used to confirm the anti tumor activity of the virus in MCTSs. Following infection, spheroids can be checked for the expression of aforementioned markers.

6. PLOS authors have the option to publish the peer review history of their article (what does this mean?). If published, this will include your full peer review and any attached files.

Reviewer #1: No

Reviewer #2: No

---

## [Author Response · Author response to Decision Letter 0]

9 Apr 2020

Dear Dr. Pizzo,

Thank you for your positive consideration of our manuscript by Lee et al. entitled, “Real-time monitoring of oncolytic VSV properties in a novel in vitro microphysiological system containing 3D multicellular tumor spheroids” (Manuscript ID PONE-D-19-34362)

The manuscript has been thoroughly revised to reflect the reviewers’ comments, including additional experiments. Here, we re-submit our revised manuscript while addressing all the issues raised on a point-by-point basis. Please find all the responses in this letter, which also provides identifying page and line numbers when referring to specific parts of the manuscript for your convenience. 

Once again, we sincerely appreciate your thoughtful consideration and look forward to hearing from you soon. 

Best regards,

Heuiran Lee, Ph.D.

Reviewer #1: The authors previously developed a microphysiological system (MPS) which features a vessel-like channel that can be lined with endothelial cells and a total of 29 side chambers which can be populated with various cell types. In the current study they generated spheroids in the side channels by infusing A549 cancer cells and MRC5 fibroblasts, then lined the vessel-like channels with HUVECs. They infused VSV-GFP into the vessel-like channels and were able to show that the virus infected only the A549 cells, not the MRC5 cells in the spheroids. This result was expected since the MRC5 cells were resistant to the virus in 2D culture and the A549 cells are susceptible. Interestingly the HUVECs did not get infected in the MPS, but were susceptible to virus infection in 2D culture. The authors speculate based their analysis of IFNb release from VSV infected A549, MRC5 and HUVEC cells, and the protection of those cells from VSV infection upon IFNb exposure, that the release of IFNb by virus-exposed MRC5 cells in the MPS is able to render the HUVEC resistant to infection.

The MPS system colonized with multicellular spheroids does offer an interesting in vitro test system for the study of oncolytic viruses that might well catch on as an alternative to mouse xenograft models if it were widely available. The ability to form multicellular tumor spheroids in side chambers accessible only via a vessel-like channel is appealing and allows for detailed analyses of the dynamical aspects of virus interaction with microvessels, tumor cells and fibroblasts in a controllable system. However, there are significant weaknesses with the current manuscript which, in the opinion of this reviewer, should be addressed prior to publication:

Ans: 

We thank you for your thoughtful consideration of our manuscript. Your comments and concerns raised are addressed below on a point-by-point basis. The manuscript has been thoroughly revised, including additional experiments. 

Rev-1 Q1: The authors state that they perfuse their system with VSV-GFP using a virus moi (multiplicity of infection) of 5.0. But what does that mean? First, they do not give details of the cell substrate on which they titrate their virus stock. Second, it is not clear how they determine the "moi" for the MPS when they do not know how many cells are present in the MPS that they are perfusing. This aspect of the paper is confusing and needs to be clarified.

Ans: Thank you for your careful reading of our work. In order to ensure the consistent infection of oVSV-GFP in this study, the stock virus titer was quantified by TCID50 indicating how many infectious virus particles are present. This information is included in the revised M&M (page 6, lines 120-122). 1 MOI represents 1 infectious virus particle per cell as described in the updated M&M section (page 6, lines 122-123). 

In MCTSs, MOI was determined based on the total number of cells (3 x 105 cells/chip) in MPS where 1 x 105 cells were identified identically from A549, MRC5 and HUVEC cells, respectively. Under this infection condition, consistent results were achieved in MPS. The above information is now found in the M&M section (page 7, 138-143; page 7-8, 147-151), the results (page 11, lines 234-235), and the legend (page 12, lines 258-260) of our updated manuscript.

Rev-1 Q2: The vessel-like channels in the MPS are supposedly lined by HUVEC cells but there are no images provided to confirm this, nor to show whether the HUVEC lining is indeed sufficiently comprehensive to form vessel-like channels. This is a critical aspect of the system and should be addressed with additional data. Without a direct visualization of the HUVEC lining cells, and an assessment of their GFP expression post virus infusion, the claim that the HUVEC cells resident in the MPS are resistant to virus infection remains unproven.

Ans: The data directly visualizing HUVEC lining cells were not included because the characteristics of MPS designed in-house, including vessel-like tubular structure with HUVEC cells, had recently been published (Fibroblast-associated tumor microenvironment induces vascular structure-networked tumouroid. Scientific reports. 2018; 8:2365) and this tubular structure was consistently observed when performing this study. However, we agree that the data showing HUVEC linings is necessary to strengthen our conclusions and appreciate your valuable suggestion. We therefore performed an additional experiment to generate MCTSs in MPS where HUVEC cells were fluorescently labeled with red PKH26 as described in the M&M. HUVEC lining in channels of MPS was clearly observed under a fluorescence microscope and the fluorescence image along with the LM image were included in the revised Figures as Fig. 1D. 

The contents have been also modified in the M&M (page 5, lines 101-102), the results (page 10, lines 205-206), and the legend (page 10, line 213). 

Rev-1 Q3: The authors state that there are 29 side chambers in their MPS but do not indicate how they captured data from all of these chambers and do not provide any statistical backup for the conclusions they have drawn.

Ans: We agree with your valuable comment and apologize for not describing our data properly. Detailed information on how data was captured from the chambers and respective statistical significance has been added to the revised manuscript whenever necessary, including conclusions. 

Rev-1 Q4: The hypothesis that HUVEC cells lining the MPS channels are rendered resistant to VSV-GFP by exposure to IFNb from MRC5 cells could easily be tested by rerunning the experiment using A549-only spheroids that do not contain MRC5 cells. Data from such an experiment should be included in the manuscript.

Ans: Thank you for thoughtful suggestion. As shown below, the GFP signal from oVSV in HUVEC lining was actually observed even at 8 h p.i. when MPS was prepared with HUVEC cells alone. In contrast, GFP signals were detected exclusively in the tumor spheroid when the virus was introduced into MPS containing all three cell lines.

Since this data is actually part of Fig. 5C, the original Fig. 5C was modified by adding a GFP image panel (the figure below) and corresponding texts, such as M&M (page 7, lines 145-146), results (page 14, lines 294-300) and legend (page 14, lines 304-307) have been also rewritten. In Fig. 5C, the effect of fluid flow on the expression of low density lipoprotein receptor (LDLR), the main receptor of oVSV, was investigated in HUVEC cells after oVSV-GFP treatment. The study was conducted using MPS composed of HUVEC cells alone. In this situation, GFP signals were substantially observed in HUVEC cells compared to MPS with all three cell lines (Fig. 4) and this phenomenon was not dependent on fluidic flow. Consistent LDLR expression levels were also identified regardless of fluidic flow, further suggesting that difference in HUVEC sensitivity between MPS and 2D system is not related to LDLR expression.

Previous studies have shown that IFN-β secreted from primary cells induces anti-viral activities in other nearby cells through paracrine signaling. The amount of IFN-β from MPS eluate was measured. Alhough not statistically significant, The IFN-β level was found to be 5.53 ± 3.94 pg/ml at 14 h p.i. (n=3). We thus believe that with the IFN-β analysis of Fig. 5B, the correlation between the initial IFN-β secretion by neighboring MRC-5 cells and HUVEC's resistance to oVSV in MPS could be suggested.

Related references: 

- Akira S, Uematsu S, Takeuchi O. Pathogen recognition and innate immunity. Cell. 2006;124(4):783–801

- Voigt EA, Swick A, Yin J. Rapid induction and persistence of paracrine-induced cellular antiviral states arrest viral infection spread in A549 cells. Virology. 2.16; 496:59-66

- Kotredes KP, Thomas B, Gamero AM. The protective role of type I interferons in the gastrointestinal tract. Frontiers in immunology. 2017; 8:410

Reviewer #2: 

Authors have investigated the potential of their recently developed microphysiological system (MPS) that mimics the in vivo tumor microenvironment and serves as culture space for 3D multicellular tumor spheroids (MCTS). This system was used as a model to check the oncolytic potential of Vesicular Stomatitis virus and concluded that this system can be used as a first line of monitoring system to validate any oncolytic virus efficacy. It is a helpful model system for evaluation of anticancer therapies including virotherapy. 

It is a good piece of work; however it was not presented very well. Methodology and results need to be written clearly. At many places the clarity is missing. Experiment description in the legends needs lot of improvement. 

Ans: We appreciate your positive consideration and valuable comments on our manuscript. With your comments in mind, we have thoroughly revised the manuscript by resolving all the issues presented below.

Rev-2 Q1: In Figure 1C, A549 cells stained green and MRC5 red. But why is it bright yellow at the interface of these two cell populations? Is it due to overlap? 

Ans: Thank you for reading our manuscript carefully. A549 cells were actually fluorescently stained green and MRC5 cells were red in MCTSs (Fig 1C). So, as you think, it appears yellow if you examine merged images in areas where two different types of cells overlap in 3D MCTSs. This information has been added to the revised manuscript (page 10, lines 202-205) to help the reader's understanding. 

Rev-2 Q2: Reference to the “effective replication of oncolytic VSV expressing GFP in A549cells” under result section, related Fig 2C, oVSV effects were shown at 5h and 50h. It would be better if 25h data of GFP expression and PI staining is also incorporated in the image panel. In this experiment, CPE/GFP expression/cell death was measured up to 50h post infection; but the negative control (mock infected) was not included. How was it confirmed without a negative control that the noted cell death was virus induced? 

Ans: As recommended, the 25H data is also incorporated in the image panel, fore Fig. 2. Unlike the 5H period, green signals indicating active oVSV-GFP replication at 25H where the growth of the virus reaches its peak was easily observed. Upon virus introduction, the proliferation of A549 human cancer cells, which has a doubling time of about one day, was not observed at both 25H and 50H under LM compared to 5H p.i. In addition, we performed a mock-treated negative control to ensure the integrity of A549 cells during the entire experimental time. As expected, untreated negative cells continue to grow to 50H as shown below. There were no signs of cytopathic effect as determined by LM and PI analysis.

Conversely, PI analysis marked cell death was observed when oVSV-GFP was actively replicated (Fig. 2B & the revised Fig. 2D). Taken together, the data indicate that the cytopathic effect is indeed driven by oVSV replication, which is included in the modified Fig. 2. The results (page 10-11, lines 217-222) and legend (page 11, lines 227-230) have been updated accordingly.

Rev-2 Q3: In Figure 3E, quantification of viral genome copies was done to analyse oVSV-GFP replication in MCTS eluates. How was it done? Was it the amplification of VSV –G gene by quantitative PCR? Please clarify in the text and legend. In this experiment as well, negative control was missing. 

Ans: We did utilize VSV-G specific primers for PCR reaction in the amplification of the RT-qPCR viral genome. The details can be found in the modified M&M (page 8, lines 162-168). In this experiment, we also performed RT-qPCR using a mock-treated negative control to ensure VSV-G RNA genome-specific reactivity as shown below. 

Since the mock-treated eluate sample has no VSV, no positive signals were detected in real-time thermal cycler regardless of RT reaction. On the contrary, large amounts of viral genomic copies were readily identified when RT reaction was performed prior to qPCR in the eluates from the viral infection group. We clarified this finding by rewriting the results (page 11, lines 238-239, page 12, lines 242) and the legend (page 12-13, lines 263-265).

Rev-2 Q4: What was the rationale behind using different MOIs, 1 and 5 for infections in 2D cultures and MPS? 

Ans: As described in the revised M&M (page 6, lines 122-123), the cell lines in 2D culture were infected with oVSV-GFP at 1 MOI to efficiently incorporate viruses into cells. Because viruses could not effectively reach the cells particularly located inside due to three-dimensional architecture of MCTS, MPS was treated with 5 MOI. MOI was determined in MPS (page 7-8. lines 147-151) based on the total number of cells (3 x 105 cells/chip), for consistent results.

Rev-2 Q5: From the methodology section given under ‘quantification of secreted human interferon beta (IFNβ) after VSV-GFP infection’ it is clear that IFNβ response was measured in all three types of cells following oVSV-GFP infection in 2D cultures and eluates collected from infected MPS by ELISA. But the related result and Figure/ figure legend are confusing. Please check the result and the figure related to this experiment and write more in detail. 

Ans: To clarify IFN-β secretion was monitored from the culture medium of each cell after oVSV-GFP treatment in 2D culture, The information on 3D eluates has been removed from M&M. We also carefully amended the M&M (page 9, lines 175-179), results (page 13-14, lines 285-290), and legend (page 14, lines 303-304). 

Rev-2 Q6: To check the differential replication of oVSV-GFP in the 3D MCTS integrated MPS, A549 cells, MRC5 cells were labeled with fluorescence tracker. Figure 4A and 4B, both the cells; A549 and MRC5 were stained red. How were these two cell populations separated to check the expression oVSV-GFP through red and green merge? Are 4A and 4B images from the same spheroid? Clarify and also mention in the text and legends. In the figure 1, A549 cells were stained green. 

Ans: Thank you for the thoughtful comments. Staining cells with a green tracker is not practical because the green signal from cells stained with green cannot be distinguished from GFP signals that appear following oVSV-GFP replication in the infected cells. In other words, when cells were infected with oVSV-GFP, A549 and MRC5 constituting MCTSs could not be stained simultaneously in two colors, green and red. 

Therefore, where MCTSs were treated with oVSV-GFP, A549 or MRC5 (Fig. 4, 4A, and 4B, respectively) cells stained with red were utilized. This would cause the red and green to overlap and turn yellow if the virus replicates in the labeled cell, whereas the red and green would not overlap if the virus did not multiply in the labeled cell. The images in Fig. 4A and Fig. 4B are not from the same MCTSs, and we revised the M&M (page 5, lines 101-102), results (page 12, lines 243-248), and legends (page 13, lines 269-274) to clarify this issue. 

Unlike Fig 4, we illustrated the mock-treated MCTS in Fig. 1, as we were able to stain A549 and MRC5 cells in two distinct colors. The corresponding content has been rewritten and can also be found in the answer to your first comment. 

Rev-2 Q7: There is a lot of room for improvement in discussion section. One sentence in the discussion from lines 326 to 329 is the exact repetition of a sentence under result section from lines 281 to 284. Improve the discussion. 

Ans: Yes, we fully agree with your comments and thank you again. Repeated sentences have been rewritten in the revised version of the manuscript (page 13-14, lines 285-290; page 15-16, lines 330-339) and the discussion section has been substantially modified to improve the quality of the manuscript.

Rev-2 Q8: The intensity of GFP expression in A549 cells does indicate the active replication of oVSV-GFP virus but not the anti tumor activity. Experiment of Live/Dead cell staining following virus infection does not commend the anti-tumor activity of oncolytic virus used. An anti tumor/ anti proliferative marker may be used to confirm the anti tumor activity of the virus in MCTSs. Following infection, spheroids can be checked for the expression of aforementioned markers. 

Ans: Thank you for the thoughtful opinion. As mentioned, the intensity of GFP expression in A549 cells does not directly indicate anti-tumor activity. However, evidence for the close correlation between oVSV replication and anti-tumor activity by monitoring oVSV-GFP replication, GFP signaling, and cytopathic effects after virus treatment in A549 cells is provided in Fig. 2. Cytopathic effects in A549 cells were determined by LM and PI staining. Anti-tumor activity in MCTSs was investigated by PI staining after oVSV-GFP treatment using tumor spheroids unlabeled with a fluorescent dye (Fig. 4C). The PI-positive dead cell region was perfectively merged with the GFP-positive region, indicating that the GFP-positive A549 cells, but not the MRC5 cells or HUVECs, were dead. Additionally, the size of infected MCTSs gradually decreased, while that of uninfected control MCTSs increased (Fig. 4D, 4E). Therefore, we believe that the results mentioned above have adequately provided evidence to confirm the anti-tumor activity of the virus even without further confirmation using anti-proliferation markers. This content has been added to the results (page 11, lines 219-222 & page 12, lines 248-254) and legend (page 11, lines 227-230 & page 13, lines 274-278).

---

## [Decision Letter · Decision Letter 1]

15 Jun 2020

Real-time monitoring of oncolytic VSV properties in a novel in vitro microphysiological system containing 3D multicellular tumor spheroids

PONE-D-19-34362R1

Dear Dr. Lee,

We’re pleased to inform you that your manuscript has been judged scientifically suitable for publication and will be formally accepted for publication once it meets all outstanding technical requirements.

Kind regards,

Salvatore V Pizzo

Academic Editor

PLOS ONE

Additional Editor Comments (optional):

Reviewers' comments:

Reviewer's Responses to Questions

**Comments to the Author**

1. If the authors have adequately addressed your comments raised in a previous round of review and you feel that this manuscript is now acceptable for publication, you may indicate that here to bypass the “Comments to the Author” section, enter your conflict of interest statement in the “Confidential to Editor” section, and submit your "Accept" recommendation.

Reviewer #2: All comments have been addressed

2. Is the manuscript technically sound, and do the data support the conclusions?

Reviewer #2: Yes

3. Has the statistical analysis been performed appropriately and rigorously? 

Reviewer #2: Yes

4. Have the authors made all data underlying the findings in their manuscript fully available?

Reviewer #2: Yes

5. Is the manuscript presented in an intelligible fashion and written in standard English?

Reviewer #2: (No Response)

6. Review Comments to the Author

Reviewer #2: Authors have addressed all the questions adequately, also incorporated suggested information and experiments. However, following are couple of things to be considered.

Rev-2 Q4: What was the rationale behind using different MOIs, 1 and 5 for infections in 2D cultures and

MPS?

Ans: As described in the revised M&M (page 6, lines 122-123), the cell lines in 2D culture were infected

with oVSV-GFP at 1 MOI to efficiently incorporate viruses into cells. Because viruses could not effectively reach the cells particularly located inside due to three-dimensional architecture of MCTS, MPS was treated with 5 MOI. MOI was determined in MPS (page 7-8. lines 147-151) based on the total number of cells (3x 105 cells/chip), for consistent results.

Comment 1: The above explanation may be incorporated either in methods or in the relevant results section.

Comment 2: In the manuscript, at some places “in vitro and in vivo” words are written in italics (in vitro/in vivo) and not everywhere. Make it uniform.

7. PLOS authors have the option to publish the peer review history of their article (what does this mean?). If published, this will include your full peer review and any attached files.

Reviewer #2: No